# The Link between Cannabis Use, Immune System, and Viral Infections

**DOI:** 10.3390/v13061099

**Published:** 2021-06-09

**Authors:** Sanjay B. Maggirwar, Jag H. Khalsa

**Affiliations:** 1Department of Microbiology, Immunology and Tropical Medicine, School of Medicine and Health Sciences, The George Washington University, Washington, DC 20037, USA; jkhalsa@yahoo.com; 2Medical Consequences of Drug Abuse and Infections Branch, National Institute on Drug Abuse, National Institutes of Health, Bethesda, MD 20852, USA

**Keywords:** cannabis, marijuana, cannabinoids, immune system, infections, HIV, HCV, HTLV-I/II

## Abstract

Cannabis continues to be the most used drug in the world today. Research shows that cannabis use is associated with a wide range of adverse health consequences that may involve almost every physiological and biochemical system including respiratory/pulmonary complications such as chronic cough and emphysema, impairment of immune function, and increased risk of acquiring or transmitting viral infections such as HIV, HCV, and others. The review of published research shows that cannabis use may impair immune function in many instances and thereby exerts an impact on viral infections including human immune deficiency virus (HIV), hepatitis C infection (HCV), and human T-cell lymphotropic type I and II virus (HTLV-I/II). The need for more research is also highlighted in the areas of long-term effects of cannabis use on pulmonary/respiratory diseases, immune dysfunction and the risk of infection transmission, and the molecular/genetic basis of immune dysfunction in chronic cannabis users.

## 1. Introduction

Today, marijuana (cannabis) is the most frequently used drug in the world, with over 188 million users, or ~2.5% of the population that is 15–64 years of age [1]. In the United States, the percentage of people aged 12 or above who were past year marijuana users increased from 11.0 percent (or 25.8 million people) in 2002 to 17.5 percent (or 48.2 million people) in 2019 [2]. Approximately 2 to 3 million new users of marijuana are added each year with about 1.1% becoming clinically dependent on it [3]. According to a recent annual survey of high school students, known as Monitoring the Future (MTF), the total annual marijuana prevalence rose by a significant 1.3% to 23.9% in 2017 with prevalence of 10%, 26%, and 37% in 8th, 10th and 12th graders, respectively, whereas daily marijuana use remained at 1%, 3%, and 6%, respectively [4]. In addition to problematic cannabis use, an estimated 38 million people are living with human immune deficiency virus infection (HIV [5]), 170 million people with hepatitis C virus (HCV [6]) infection, 10–20 million people with human T-lymphotropic virus type 1 (HTLV-1 [7]), and an estimated 161 million people have been infected with coronoavirus, SARS-Cov2 [8], in the world. Cannabis use is associated with a wide range of adverse economic, social, psychosocial and health consequences. The psychosocial consequences of marijuana use—such as dropping out of school, poor school performance, and antisocial and other behaviors among youth—have been the subjects of many reviews/publications. The health consequences of cannabis use involve almost all physiological and biochemical systems including the immune, cardiopulmonary/respiratory, hepatic, renal, endocrine, reproductive, and central nervous systems, as well as genetics and general health [9,10,11,12]. This review presents current research on the impact of cannabis use on the immune system which, in turn, may lead to increased risk of acquisition and transmission of viral infections including HIV, HCV, HTLV-I/II and SARS-CoV2.

## 2. Methods

To prepare this review, we used the NIH PubMed database and Google Scholar, by using key words including cannabis, cannabinoids, tetrahydrocannabinol, THC, cannabidiol, CBD, marijuana, immune system, viral infections, HIV, hepatitis C infection (HCV) and HTLV-I/II, and identified, retrieved and reviewed the papers published in the English language that are relevant to the topic of a link between cannabis, the immune system, and viral infections.

## 3. Cannabis and Immune System

The cannabis sativa plant contains 560 chemicals, of which 104 are known as cannabinoids [13]. The most studied cannabinoids are Δ^9^-tetrahydrocannabinol (THC), which produces the majority of its psychopharmacological and other effects through two cannabinoid receptors, CB1 (localized mainly in the brain) and CB2 (localized mainly in the periphery), and cannabidiol (CBD), a non-psychoactive cannabinoid. The endogenous cannabinoids, known as the endocannabinoid system (ECS), consist of the endogenous lipid ligands N-arachidonoylethanolamine (anandamide; AEA) and 2-arachidonylglycerol (2-AG), their biosynthetic and degradative enzymes, and the CB1 and CB2 receptors that they activate.

Endocannabinoids are present on immune cells such as monocytes, macrophages, basophils, lymphocytes, and dendritic cells and are enzymatically produced and released “on demand” in a similar fashion as the eicosanoids. Cannabidiol (CBD) and cannabinol (CBN) can alter the functional activities of the immune system [14]. Cannabis and its active cannabinoids act as immune-modulating agents, affecting T-cells, B-cells, monocytes, and microglia, causing an overall reduction in proinflammatory cytokine expression and an increase in anti-inflammatory cytokines. The role of ECS, CB1, CB2 and cannabinoids in various physiological systems, including immunity and different pathologies, has been well discussed by Olah et al. [15], and Almogi-Hazan and Or [16]. More recently, De Silva et al. [17] suggested the role of neuroimmune activation and the subsequent increase in 18-kDa translocator protein (TSPO) levels in cannabis users. In a cross-sectional study of 24 long-term cannabis users and 27 non-cannabis using controls, by using PET scans, they measured the total distribution volume across regions of interest, stress and anxiety, peripheral measures of inflammatory cytokines and C-reactive protein levels, and found that cannabis users had higher total distribution volume, thus implicating it in the development of cannabis use disorder. Greater TSPO levels in the brain were found to be associated with stress and anxiety and with higher circulating C-reactive protein levels in cannabis users.

Cannabidiol may serve as an immune suppressor via the suppression of activation of various immune cell types, the induction of apoptosis, and the promotion of regulatory cells, which, in turn, control other immune cell targets [18]. Although cannabis impairs cell-mediated and humoral immunity in animal models and decreases resistance to bacterial and viral infections [19], there is no conclusive evidence to suggest that cannabis use impairs immune function, as measured by the number of T cell lymphocytes, B cell lymphocytes, macrophages, or levels of immunoglobulin in humans [20,21,22]. On the other hand, and more importantly, the chronic heavy smoking of cannabis weakens the immune system, leading to increased symptoms of chronic bronchitis, coughing, production of sputum, and wheezing [23,24,25], and impairs the pulmonary function, pulmonary responsiveness, and bronchial cell characteristics in cannabis-only smokers. Tashkin et al. [26] further show that chronic cannabis smoking is associated with poorer lung function and greater abnormalities in the large airways of marijuana smokers than in nonsmokers, and a greater rate of decline in respiratory function among marijuana-only smokers than in tobacco-only smokers [27]. Both studies showed that long-term smoking of marijuana increased bronchitis symptoms. Cannabis smokers also show significantly higher levels of cytologic components in their sputum when compared with sputum from tobacco smokers [28]. Thus, marijuana smoking may predispose individuals to pulmonary infection, especially among patients whose immune defenses are already compromised by HIV infection and/or cancer-related chemotherapy [26]. They reported that THC produced a concentration-dependent reduction in T cell proliferation and IFN-γ production via a CB2 receptor-dependent pathway, and at the level of gene expression, THC increased the expression of Th1 cytokines (IFN-γ/IL-2) and reduced the expression of Th2 cytokines (IL-4/IL-5). However, they caution that the suppression of cell-mediated immunity by Δ^9^-THC may place marijuana smokers at risk for infection or cancer. Caiaffa and colleagues [29] reported almost four times higher incidence of bacterial pneumonia in HIV-seropositive subjects than among HIV-negative subjects, and smoking illicit drugs (marijuana, cocaine, or crack) had the strongest effect on the risk of bacterial pneumonia among HIV-seropositive IDUs with a previous history of Peumocystic carinii pneumonia (PCP). Smoking cannabis and drinking alcohol also significantly increase the IL-6 cytokine and toll-like receptors, TLR5, TLR7 and TLR9s, that recognize airborne microbes and initiate the inflammatory cytokine response, in cannabis users’ bronchoalveolar lavage and airway epithelial brushing, suggesting cannabis-related effects on the pulmonary innate immunity promoting airway inflammation [30]. On the other hand, in a well-designed in-patient study of HIV-infected patients on anti-retroviral therapy that smoked well characterized NIDA-supplied marijuana cigarettes with known concentrations of THC, Brendt and colleagues [31] found no significant changes in naive/memory cells, activated lymphocytes, B cells, or NK cell numbers that could be directly attributed to the administration of cannabis in HIV-infected patients on anti-retroviral therapy. In the case of cannabis use during pregnancy, since cannabinoids cross the placental barrier, it would negatively impact the fetal immune system, primarily via G-protein coupled cannabinoid CB1 and CB2 receptors, and thus could dysregulate the innate and adaptive immune system of the developing fetus and offspring, potentially leading to weakened immune defenses against infections and cancer later in life [32].

Many inflammatory conditions are associated with dysfunction of the immune system. Cannabis has been used for centuries as a medicine in the treatment of a variety of inflammatory disorders including rheumatic arthritis (RA), gastrointestinal (GI) diseases such as Crohn’s disease (CD) and inflammatory bowel disease (IBD), and other GI problems such as anorexia, emesis, abdominal pain, diarrhea, and diabetic gastroparesis. Furthermore, in the endocannabinoid system, anandamide, an endogenous intestinal cannabinoid, and its receptor, CB2, regulate the immune system in the gut and control the appetite and energy balance by engagement of the enteric nervous system [33]. The activated ECS reduces gut motility, intestinal secretion, and epithelial permeability, and induces inflammatory leukocyte recruitment and immune modulation through the cannabinoid receptors present in the enteric nervous and immune systems. Therefore, attempts have been made to investigate the medicinal properties of cannabinoids, particularly for treating pancreatitis, hepatitis, and inflammatory bowel disease (IBD). Further, since cannabinoids decrease the production of cytokines and immune mobilization via activation of CB2 receptors, it has been suggested that cannabinoids could be used to treat inflammatory conditions such as rheumatoid arthritis [34], osteoarthritis, fibromyalgia, systemic sclerosis, and juvenile idiopathic arthritis [35]. However, the current evidence is insufficient to support the recommendation of cannabinoids for treating rheumatic diseases. In the case of GI inflammatory disorders such as Crohn’s disease, in a small observational study of 30 patients with CD, medicinal use of cannabis improved the disease and reduced the use of other medications [36]. Cannabis also decreased the disease in 10 of the 11 patients compared to 4 of the 10 controls, and led to a complete remission in 5 of the 11 cannabis group patients and one of the 10 controls [37]. However, in a trial by Naftali et al. [38], a low dose of CBD was safe but ineffective in patients with CD, suggesting that further research to support the use of CBD for the treatment of inflammatory diseases is needed. Picardo and colleagues [39] also suggest that even though small studies show the effectiveness of cannabis in the treatment of IBD and other inflammatory disorders, large clinical trials are still needed to support the use of cannabis in the treatment of inflammatory disorders.

Continuing on the subject of cannabis as medicine, there have been claims that either smoked or ingested cannabis containing the psychoactive component THC, and those that are natural or synthetic in origin (dronabinol), improves the appetites of people with AIDS, increases weight gain and lifts mood, thereby improving the quality of life. In a double-blind cross-over study [40], compared to placebo, cannabis administration was associated with significant increases in plasma levels of appetite controlling hormones, ghrelin and leptin, but did not significantly influence insulin levels, suggesting that the modulation of appetite hormones is mediated through endogenous cannabinoid receptors, independent of glucose metabolism. In an elegant NIDA-funded randomized, cross-over, double-blind, placebo-controlled study, Farokhnia et al. [41] investigated the effects of cannabis administration, via different routes, on peripheral concentrations of appetitive and metabolic hormones in a sample of cannabis users. Twenty participants underwent four experimental sessions during which oral cannabis, smoked cannabis, vaporized cannabis, or placebo was administered. Active compounds contained 6.9 ± 0.95% (~50.6 mg) THC. Repeated blood samples were obtained, and the following endocrine markers were measured: total ghrelin, acyl-ghrelin, leptin, glucagon-like peptide-1 (GLP-1), and insulin. Results showed a significant drug main effect (*p* = 0.001), as well as a significant drug × time-point interaction effect (*p* = 0.01) on insulin. The spike in blood insulin concentrations observed under the placebo condition (probably due to the intake of a brownie) was blunted by cannabis administration. A significant drug main effect (*p* = 0.001), as well as a trend-level drug × time-point interaction effect (*p* = 0.08) was also detected for GLP-1, suggesting that GLP-1 concentrations were lower under cannabis, compared to the placebo condition. Finally, a significant drug main effect (*p* = 0.01) was found for total ghrelin, suggesting that total ghrelin concentrations during the oral cannabis session were higher than the smoked and vaporized cannabis sessions. The investigators concluded that cannabis administration modulated blood concentrations of some appetitive and metabolic hormones, chiefly insulin, in cannabis users.

## 4. Cannabis and Infections

People living with viral infections use many legal and illegal substances, including the most used substance, cannabis. There have been debates as to whether cannabis can be used as a medicine in people with infections including HIV. Let us examine whether cannabis use is associated with positive or negative effects in people living with viral infections. With regard to its negative effects, there is a clear paucity of data that would support the notion that cannabis is immunotoxic or that it increases the risk of exacerbating other bacterial or viral diseases in cannabis users. Recent studies [42,43,44,45] suggested that cannabis and other drugs of abuse may modulate the immune system by acting on specific receptors on immune cells, decrease immunity, and thereby increase susceptibility to infections including HIV and HIV-associated opportunistic infections. Earlier research in humans and macaques suggested that cannabis may have an impact on plasma viral load [46]. However, later research suggested that cannabis use may be potentially beneficial in people with HIV infection [47], where the investigators studied the impact of cannabis use on peripheral immune cell frequency, activation, and function in 198 HIV-infected, antiretroviral-treated individuals, and found that heavy cannabis users had decreased frequencies of intermediate and nonclassical monocyte subsets, as well as decreased frequencies of interleukin 23- and tumor necrosis factor-α-producing antigen-presenting cells, compared to frequencies of these cells in non-cannabis-using individuals, suggesting beneficial effects of cannabis. Although the clinical implications of these findings are unclear, data suggest that cannabis use is associated with a potentially beneficial reduction in systemic inflammation and immune activation in the context of antiretroviral-treated HIV infection. Medicinal or recreational cannabis use increases susceptibility to bacterial, viral, parasitic and fungal infections, possibly via the modulation of the immune system [19]. In a prospective study of a cohort of HIV-infected homosexual men, cannabis use was not associated with increased risk of progression to AIDS [48]. Kaslow and colleagues [49] also conducted a prospective study of the progression of HIV infection to AIDS among HIV+ men in a cohort of 4954 homosexual and bisexual men. The participants were asked about the recency of marijuana use (within the previous 2 years, 6 months and in the last 7 days; and whether used daily, weekly, monthly, or less often). Regardless of frequency of use, marijuana use neither increased the rate of progression to AIDS, nor impaired the immune function, in HIV+ men. Thus, although HIV-infected persons have been advised to avoid marijuana, this advice appears to be reasonable as a general health precaution. The fact that Marinol (dronabinol, contains synthetic THC), which has been approved by the FDA for the treatment of anorexia associated with weight loss in patients with AIDS and chemotherapy-associated nausea and vomiting, also does not impair the immune system significantly and does not exacerbate bacterial or viral infections, further suggests that cannabis use, most likely, does not adversely impact on HIV disease progression. Thus, although HIV-infected persons have been advised to avoid marijuana, this advice appears to be reasonable as a general health precaution.

In the context of aging and HIV, cannabis use may also exert beneficial effects due to its anti-inflammatory properties. Watson et al. [50] examined, in a cohort of 679 HIV-positive and 273 HIV-negative people (18–79 years old), the independent and interactive effects of HIV and cannabis on neurocognitive impairment (NCI) and the potential moderation of these effects by age. They found a significant interaction between HIV and cannabis (*p* = 0.02). Among people living with HIV (PLHIV), cannabis was associated with a lower proportion of NCI but not among HIV- negative individuals (*p* = 0.40), with no impact of age, suggesting that cannabis exposure is linked to lower odds of NCI in the context of HIV, possibly via the anti-inflammatory effect of cannabis, which may be particularly important for PLHIV [50].

Does cannabis use affect morbidity or mortality in people with viral infections? The evidence for substantial effects on morbidity and mortality is currently limited. Data from only one relatively small study (*n* = 139, of which only 88 subjects were evaluable), conducted in the period before access to antiretroviral therapy (ART), showed that the mean weight gain in the dronabinol group was only 0.1 kg compared with a loss of 0.4 kg in the placebo group. Thus, although dronabinol has been approved for the treatment of AIDS-associated anorexia, evidence for the efficacy and safety of cannabis and cannabinoids for people living with HIV/AIDS is still lacking. Currently, there are no studies of chronic cannabis use to show a sustained effect on AIDS-related morbidity and mortality and safety in patients on antiretroviral therapy. Whether the available evidence is sufficient to justify a wide-ranging revisiting of the regulatory practice of medicines remains unclear [51]. Long-term data, showing a sustained effect on AIDS-related morbidity and mortality and safety in patients on effective antiretroviral therapy, has yet to be presented. In a prospective study (ACCESS study in Vancouver, Canada), a cohort of 874 HIV-infected people who used drugs was studied for the impact of cannabis on ART care and engagement. A multivariate analysis showed that daily use of cannabis did not produce lower odds of ART care including during periods of binge alcohol drinking. There was no statistically significant impact of daily use of cannabis on the likelihood of ART care among ART-exposed HIV-positive persons who used drugs [52]. Based on data from cannabis users enrolled in the Ontario HIV Treatment Network cohort (*n* = 763; 67% white, 88% male, 68% gay, medium income in the range of $40,000–$50,000), Wardell et al. [53] found that there was a greater frequency of cannabis and alcohol use. The medicinal cannabis users reported more frequent use and less alcohol use on average than recreational cannabis users. Cannabis use and alcohol use were positively associated with each other over time among PLHIV, although this association was specific to those using cannabis for recreational reasons. Since alcohol use in this population poses significant health risks, it was suggested that more research be conducted to determine the link between alcohol and cannabis use, particularly in light of recent changes to cannabis regulations [53].

Whether or not there is clear evidence to show that cannabis use impacts on treatment adherence/management among people with viral infections remains unclear. In a cross-sectional observational study of 180 HIV+ subjects (78.3% male), chronic cannabis users (cannabis-dependents) reported lower medical adherence (measured via pill count, viral load and CD4 count) and greater HIV symptoms/ART side effects than the non-dependent users or non-users of cannabis, suggesting a clinical need to address dependent cannabis use among those prescribed ART [54]. In another study of 703 HIV+ participants [55], 33.2% reported using marijuana in the past 3 months. Among the marijuana users, 21.8% reported using marijuana for therapeutic purposes (improve appetite, induce sleep, relieve nausea/vomiting, or relieve anxiety/depression) while 78.2% reported recreational use. After controlling for covariates, therapeutic use of marijuana was not associated with ART adherence (AOR = 1.19, 95% CI = 0.60–2.38, *p* = 0.602) while recreational marijuana users showed significantly greater odds of suboptimal ART adherence compared to nonusers (AOR = 1.80, 95% CI = 1.18–2.72, *p* = 0.005). The investigators suggest that continued research examining the health implications of marijuana use among adults living with HIV is important as legalization of recreational and medical marijuana proliferates in the United States. On the other hand, in a cross-sectional study of 107 HIV-positive subjects, (cannabis use, 41; non-cannabis use, 66), where cannabis use was either self-reported or ascertained by urine analysis, it was found that cannabis use had no effect on adherence to HIV therapy (52 subjects) in people living with HIV. Further, the amount of cannabis used was also not associated with measures of adherence and management. In a longitudinal study of 523 HIV+ subjects, of which 121 reported daily use of cannabis, bivariate and multivariate analyses showed that daily use of cannabis also had no effects on optimal adherence to ART (Adjusted Odds Ratio = 1.12, 95% Confidence Interval (95% CI): 0.76–1.64, *p*-value = 0.555.). The investigators suggested that cannabis may be utilized by people living with HIV for medicinal and recreational purposes without compromising effective adherence to ART [56]. Nevertheless, it would be prudent for patients to not engage in cannabis use while on antiretroviral therapy for HIV infections. In addition, cannabis has also been anecdotally used to treat common symptoms and complications including poor appetite and neuropathy in patients with HIV. Based on analysis of data from HIV-positive individuals attending a large clinic, Woolridge et al. [57] reported that up to 27% (143/523) reported using cannabis for treating symptoms; patients reported improved appetite (97%), muscle pain (94%), nausea (93%), anxiety (93%), nerve pain (90%), depression (86%), and paresthesia (85%). Although many cannabis users (47%) reported associated memory deterioration in HIV-positive outpatients, several participants reported that cannabis improved their symptom control [57]. Further, since ECS regulates homeostasis of the gut microbiome, HIV leads to alterations in the gut microbiome and gut–brain axis signaling, as well as chronic inflammation and neuroinflammation. Thus, it has been suggested that cannabis, or CBD that potentially has antioxidant and anti-inflammatory properties, may be used as an adjunct therapy in healing and restoring the gut microbiome in HIV-infected subjects [58]. However, more clinical research is needed to collect evidence that would support the use of CBD in the treatment of these inflammatory disorders. In the same vein of new avenues of cannabis research, recent developments in inflammasome research also suggest that the anti-inflammatory action of cannabinoids may be mediated, in part, by modulating inflammasome assembly and function and, thus, cannabinoids including CBD could be further developed to treat inflammatory disorders caused by viral infections including HIV and now COVID-19 [59].

Regarding chronic hepatitis C virus (HCV) infection, that is one of the most common chronic liver diseases, several risk factors such as cannabis use have been identified for the progression of liver fibrosis among these patients, but the results from epidemiological studies remain inconclusive. In an earlier study by Hezode et al. [60], cannabis use was found to be a risk factor for the progression of liver fibrosis in chronic HCV infection. In another prospective cohort study of 204 patients with chronic HCV infection, daily use of cannabis was significantly (OR, 3.2; 95% CI, 1.20–8.56, *p* = 020) associated with moderate to severe liver fibrosis (F3–6 versus F1–2) compared with occasional or never use of cannabis [61]. Cannabis use frequency (within the previous 12 months) was daily in 13.7%, occasional in 45.1%, and never in 41.2% of subjects. Fibrosis stage, assessed by the Ishak method, was F0, F1–2, and F3–6 in 27.5%, 55.4%, and 17.2% of subjects, respectively. Wijarnpreecha et al. [62] carried out a meta-analysis of three published cohort studies with a combined total of 898 patients and showed that the risk of advanced liver fibrosis among HCV-infected patients who use cannabis was higher than those who do not use cannabis, although the result did not achieve statistical significance (pooled odds ratio, 1.77; 95% CI 0.78–4.02). However, in a much larger study of 540 HIV/HCV co-infected women, of which 321 (56%) did not use THC (cannabis), 141 (25%) used THC less than weekly, 70 (12%) used weekly, and 40 (7%) used THC daily, Kelly et al. [63] showed that neither the frequency nor the amount of THC (cannabis) use had any effect on liver fibrosis in HIV/HCV co-infected women. Hepatic steatosis is also common in HIV/HCV-co-infected patients. Given the causal link between insulin resistance and steatosis, Nordman et al. [64] conducted a study of a French multi-center cohort of HIV/HCV co-infected people (ANRS CO13-HEPAVIH) to determine if cannabis use in this population had an impact on hepatic steatosis. Among study participants (*n* = 838), 40.1% had steatosis, 14% reported daily cannabis use, 11.7% regular use and 74.7% no use or occasional use (“never or sometimes”). Daily cannabis use was independently associated with a reduced prevalence of steatosis (adjusted odds ratio (95% CI) = 0.64 (0.42; 0.99); *p* = 0.046), after adjusting for body mass index, hazardous alcohol consumption and current or lifetime use of lamivudine/zidovudine. Data showed that daily cannabis use may be a protective factor against steatosis in HIV/HCV co-infected patients, suggesting a need for a clinical evaluation of cannabis-based pharmacotherapies in this population [64].

Regarding other viral infections, since cannabis smoking and vaping is associated with cerebrovascular and neurological systems and considering that smokers are more prone to viral and bacterial infection compared to non-smokers, it is possible that cannabis smoking or vaping might also exacerbate the cerebrovascular and neurological dysfunction observed in patients with SARS-Cov2 infection, which results in disease known as COVID-19 [65]. Finally, an estimated 10–20 million people may be living with human T-lymphotropic virus type I (HTLV-I) infection in the world, which is associated with slowly progressing myelopathy or tropical spastic paraparesis and HTLV-I antibodies in serum and cerebrospinal fluid [66]. Incidentally, myelopathy predominantly affects the pyramidal tracts [66] and typically presents as motor dysfunction with a variable degree of sensory dysfunction in the lower limbs and often includes sphincter and bladder disturbances [67,68]. Although illicit drug use has been reported in people with HTLV-I and II infections [69], there are no reports of cannabis use in people with HTLV-I/II infection or the use of cannabis for treating HTLV-I/II infection.

## 5. Discussion

Cannabis use is associated with a wide range of adverse social, economic and health consequences, the latter involving almost every physiological and biochemical system. However, research shows that cannabis or its constituent CBD also has some beneficial effects. Relevant to the current topic, research shows that smoking cannabis, THC, CBD and possibly other cannabinoids may alter/impair the immune system [14,15,16], activate the neuroimmune function and increase the inflammatory cytokines and C-reactive protein levels that are associated with stress and anxiety in cannabis users with cannabis use disorder [37]. CBD may also act as an immune suppressor via the suppression of activation of various immune cell types, the induction of apoptosis, and the promotion of regulatory cells, which, in turn, control other immune targets [18]. Data show that although cannabis impairs the cell-mediated and humoral immunity in rodents and decreases the resistance to bacterial and viral infections [19], there is no conclusive evidence that cannabis use impairs immune function in humans as measured by number of T cell lymphocytes, B cell lymphocytes, macrophages, or levels of immunoglobulin [20].

The pioneering observations by Tashkin and colleagues [26], and others, although conducted in the 1990s, reported important and relevant findings showing that heavy cannabis smoking weakens the immune system, resulting in a wide range of respiratory/pulmonary complications including chronic bronchitis, coughing, production of sputum, wheezing, impaired pulmonary function, pulmonary responsiveness, abnormalities in the large airways, and decline in respiratory function [27], possibly through a reduction in T-cell proliferation and IFN-gamma production, via the CB2 receptor pathway [26]. Cannabis smokers also show significantly higher levels of cytologic components in their sputum [28]. Tashkin and colleagues correctly believed that cannabis smoking may predispose individuals to pulmonary infection, especially patients whose immune defenses are already compromised by HIV infection and/or cancer and related chemotherapy and, thus, cautioned that the suppression of cell-mediated immunity by Δ^9^-THC may place marijuana smokers at risk of infection or cancer. In fact, there is a four-times-higher incidence of bacterial pneumonia in HIV-seropositive subjects with a history of peumocystic carinii pneumonia [29]. Further, in cannabis users, there were increased levels of cytokines, including IL-6, IL-8, TNFa, and IL-10, in cannabis smokers’ bronchoalveolar lavage and epithelial brushing [30]. However, in a placebo-controlled study, there were no significant changes in naive/memory cells, activated lymphocytes, B cells, or NK cell numbers in HIV-infected patients who smoked cannabis daily for 21 days, suggesting no significant effects of cannabis on immune function [31]. The effects of cannabis could be explained by the effects of THC on mitochondrial function and cellular energy that was dose-dependently reduced by THC [70]. Recently, Beji et al. [71] reviewed the available data on cannabinoids, viral infections and the role of mitochondria and concluded that cannabinoids have the potential to affect a broad range of cell types through mitochondrial modulation, be it through receptor-specific action or not, and that this pathway has a potential implication in cases of viral infections. However, a significant amount of basic and clinical research would be needed to show that cannabinoids (CBD, THC, and others) can be used to treat any of the mitochondrial disorders such as seizures, or complications of viral infections. Incidentally, CBD (Epidiolex) is approved by the FDA for the treatment of seizures associated with Lennox–Gastaut syndrome, Dravet syndrome, and tuberous sclerosis complex (TSC) in children of 1 year of age and older.

Could cannabis or CBD be used to treat any inflammatory disorder including rheumatic arthritis, Crohn’s disease, inflammatory bowel disease, and other GI problems such as anorexia, emesis, abdominal pain, diarrhea, and diabetic gastroparesis [34], or fibromyalgia, systemic sclerosis, and juvenile arthritis [35]? The currently available research does not seem to support the use of CBD for treating any of the above inflammatory disorders. However, because of the anti-inflammatory effects of cannabinoids, which seem to modulate inflammasomes [59], more systematic clinical research including randomized, double-blind, placebo-controlled trials is needed to develop CBD as a medicine for the treatment of inflammatory disorders.

Regarding cannabis and infections, earlier research in non-human primates suggested that cannabis use may increase the viral load of viruses such as HIV. However, later research showed that cannabis use does not increase the risk of HIV transmission or other bacterial or viral infections in people who smoke cannabis. Cannabis smoking also does not increase the progression of HIV to AIDS [48,49]. On the other hand, cannabis use may reduce the systemic inflammation and immune activation in HIV-infected patients on antiretroviral therapy [47]. Due to possible anti-inflammatory effects, cannabis use may also lower the odds of neurocognitive impairment in people with HIV infection [50]. Regarding the impact of cannabis on sexually transmitted infections (STIs), the data show that there were fewer STIs and lower risk of sexual engagement among HIV-infected MSMs who smoked cannabis as compared to those who did not smoke cannabis [72]. On the positive side of cannabis use, research shows that cannabis improves appetite, food intake, and metabolism, possibly via the endocannabinoid system, which, in turn, activates appetite stimulating hormones such as ghrelin and leptin [40,41].

Initially, studies by Hezode et al. [60], Ishida et al. [61], and Wijarnpreecha et al. [62] showed that cannabis use was associated with increased risk of advanced liver fibrosis among HCV-infected people. However, a study by Kelly et al. (2016) showed that cannabis use had no impact on liver fibrosis in HCV-infected patients. On the contrary, Nordman et al. [64] showed that daily cannabis use reduced the prevalence of steatosis, suggesting that daily cannabis use may be a protective factor against steatosis in HIV-HCV-co-infected patients. They suggested that additional clinical evaluation of cannabis-based pharmacotherapies in this population is needed. Regarding other viral infections such as HTLV, though other illicit drug use has been reported by people with HTLV, cannabis use among HTLV infections has not been reported. Finally, cannabis smoking or vaping may exacerbate cerebrovascular and neurological dysfunction in patients with COVID-19 [65]. Whether cannabis or CBD can be used to treat any of the COVID-related health effects remains unclear at this time [73].

As suggested below in the Future Directions section, since greater frequency of recreational cannabis and alcohol use has been reported among people with HIV infection that pose greater health risks, Wardell et al. [53] suggest more research to determine the link between alcohol and cannabis use, particularly in light of recent changes to cannabis regulations. More short- and long-term studies are also needed to determine if cannabis use would increase morbidity or mortality in people with HCV or HIV and those undergoing anti-retroviral therapy [51].

## 6. Future Directions

Based on the evidence reviewed above, additional lines of research are suggested: (i) because of variability of chemical constituents in cannabis from different sources, cannabis used in research must be of research grade and well characterized in terms of its chemical composition; (ii) basic, clinical studies, and randomized double-blind, placebo-controlled clinical trials, where appropriate, are needed to determine the effects of cannabis, or individual cannabinoids such as THC, CBD, and CBN, alone or in combination at various dose levels on immune function and viral infections; (iii) since the anti-inflammatory action of cannabinoids (CBD) may be mediated via the modulation of inflammasome assembly and function, clinical studies are needed to develop CBD as a medicine for the treatment of inflammatory conditions of the GI tract, such as Crohn’s disease, and those caused by infections, such as HIV and now SARS-Cov2; (iv) studies are needed to determine if CBD or a combination with THC or other cannabinoids can be used to treat short-and long-term complications of viral infections including HIV, HCV, and SARS-Cov2, and disease progression; (v) more short- and long-term studies are needed to determine if cannabis use would increase morbidity in people with HCV or HIV, and in HIV/HCV co-infected patients, on anti-retroviral therapy; (vi) determine if CBD can be developed to treat mitochondrial disorders; and finally (vii) in the light of current legalization of recreational and medicinal cannabis by many US states, research into the effects of alcohol and cannabis on immune function and the progression of infectious diseases is warranted.

## 7. Conclusions

Research suggests a link between cannabis, immune function, and viral infections. Cannabis use may be associated with adverse effects on immune function and, thereby, increase the risk of acquiring or transmitting infections such as HIV and HCV. However, data are not sufficiently strong to suggest that cannabis use adversely affects the progression of viral diseases. Cannabis use is also associated with adverse respiratory/pulmonary complications such as chronic cough and emphysema, and the impairment of immune function. However, it is also evident that cannabis or its constituents, including THC and CBD, have some beneficial effects such as improving appetite and food intake in patients with HIV/AIDS and positive effects in patients with hepatic steatosis. Nevertheless, as suggested above, more research is needed to study the long-term effects of cannabis use on pulmonary/respiratory diseases, immune function and the risk of infection transmission, and the molecular/genetic basis of immune dysfunction in chronic cannabis users.

## Data Availability

Data on subjects in the respective studies are available from the cited authors.

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
