# Peer review of "The Link between Cannabis Use, Immune System, and Viral Infections"

_viruses, 2021, doi:10.3390/v13061099_

Round 1

Reviewer 1 Report

Manuscript ID: 1162680

Title: The link between cannabis use, immune system, and viral infection

Viruses

This is a relevant review on the topic of cannabis use interactions and effects on the immune system and viral infection.  It hits appropriate concepts and has a defined outline. Within each of the two sections (Cannabis and Immune System; Cannabis and Infections) the authors list various studies and summarize the findings. The Discussion section summarizes additional findings but a shortcoming is the lack of discussing the findings and potentially discussing future directions necessary that can move the research field forward. Further, when presenting studies with different findings it would be nice to talk a little more about the strengths and weaknesses and caveats of each study. Overall, this article provides a review of an important and timely topic but needs to present more discussion points, interpretation, and future directions. Specific suggestions and other comments to consider:

Major Comments:

  • Presentation of variable results between studies should include a discussion on shortcomings or strengths of studies and reasons for the differences
  • In various places it is pointed out that additional research is needed as well as more systematic randomized, double-blind, placebo-controlled studies, and short- and long-term studies but no potentially future directions are discussed that are necessary to move the research field forward – it is suggested to integrate a sub-section on how to move the field forward by using the knowledge acquired and explicitly stating what needs to be focused on in the future
  • The same sentences are repeated in topic sections and the Discussion section, e.g.
    • Page 6, line 237: “Since alcohol use in this population poses significant health risks, it was suggested that more research to determine the link between alcohol and cannabis use, particularly in light of recent changes to cannabis regulations [51].”; AND page 9, line 347: “… Wardell et al. [51] suggest more research to determine the link between alcohol and cannabis use, particularly in light of recent changes to cannabis regulations.”
    • Page 5, line 199: “Thus, although HIV infected persons have been advised to avoid marijuana, this advice appears to be reasonable as a general health precaution. The fact that Marinol (dronabinol, contains synthetic THC), that has been approved by the FDA for the treatment of anorexia associated with weight loss in patients with AIDS and chemotherapy-associated nausea and vomiting, also does not impair immune system significantly and does not exacerbate bacterial or viral infections.” AND page 8, line 328: “Incidentally, Marinol (dronabinol, [synthetic THC]), approved for treating chemotherapy-induced nausea and vomiting, does not impair immune function and does not exacerbate bacterial or viral infections. Although people infected with HIV have been advised to avoid marijuana, this advice appears to be reasonable as a general health precaution.”
  • Use more transitions and "summing up" type sentences. It's useful to have an introductory sentence at the start of each paragraph that fits the paragraph into the overall train of thought. 
  • What are the unique effects of cannabis for each of the mentioned viral infections?
  • A summary graphic or summary tables would help for the reader to understand the unique effects cannabinoids (e.g. THC, CBD) have on the immune system and viral infection
  • Page 6, line 240: “Cannabis use does not adversely influence medication adherence/management among 240 adults living with HIV [52].” --> that is actually debated, different papers show otherwise, depending on how cannabis is used, medical or recreational
  • The discussion is missing a more detailed review analysis on what the difference is between the diseases outlined and the actions of these various cannabinoids

Minor Comments:

  • Page 2, line 68: “pro-inflammatory” --> should not have a hyphen; it is one word “proinflammatory”
  • Page 3, line 80: “… in turn, control other …” --> should read “… in turn, controls other …”
  • Page 5, lines 177-179: “Earlier research in humans and macaques suggested that cannabis may have an impact on plasma viral load. But later research suggests that cannabis use may be potentially beneficial in people with HIV infection.”  --> citations needed

Author Response

Briefly, we have deleted the duplicate sentences, inserted the introductory and concluding statements where appropriate, and added appropriate statements and references on cannabis and adherence to ART in patients with HIV infection. We have also corrected minor spelling errors on lines 68 and 80. 

  • Presentation of variable results between studies should include a discussion on shortcomings or strengths of studies and reasons for the differences. RESPONSE: We have mentioned, where appropriate, whether the study cited was cross-sectional or a prospective, that occasionally give different or opposing results, and where cannabis use measured was unclear. 
  • In various places it is pointed out that additional research is needed as well as more systematic randomized, double-blind, placebo-controlled studies, and short- and long-term studies but no potentially future directions are discussed that are necessary to move the research field forward – it is suggested to integrate a sub-section on how to move the field forward by using the knowledge acquired and explicitly stating what needs to be focused on in the future. RESPONSE: Thank you for suggesting, we have now included a sub-section "Future Direction" that outlines the avenues that needs to be considered in future studies.
  • The same sentences are repeated in topic sections and the Discussion section, e.g.
    • Page 6, line 237: “Since alcohol use in this population poses significant health risks, it was suggested that more research to determine the link between alcohol and cannabis use, particularly in light of recent changes to cannabis regulations [51].”; AND page 9, line 347: “… Wardell et al. [51] suggest more research to determine the link between alcohol and cannabis use, particularly in light of recent changes to cannabis regulations.”
    • Page 5, line 199: “Thus, although HIV infected persons have been advised to avoid marijuana, this advice appears to be reasonable as a general health precaution. The fact that Marinol (dronabinol, contains synthetic THC), that has been approved by the FDA for the treatment of anorexia associated with weight loss in patients with AIDS and chemotherapy-associated nausea and vomiting, also does not impair immune system significantly and does not exacerbate bacterial or viral infections.” AND page 8, line 328: “Incidentally, Marinol (dronabinol, [synthetic THC]), approved for treating chemotherapy-induced nausea and vomiting, does not impair immune function and does not exacerbate bacterial or viral infections. Although people infected with HIV have been advised to avoid marijuana, this advice appears to be reasonable as a general health precaution.” RESPONSE: The duplicate sentences were now removed.
  • Use more transitions and "summing up" type sentences. It's useful to have an introductory sentence at the start of each paragraph that fits the paragraph into the overall train of thought.  RESPONSE: Such transitional sentences are now added (thank you).
  • What are the unique effects of cannabis for each of the mentioned viral infections? RESPONSE: We believe we have addressed this as much as possible based on available literature.
  • A summary graphic or summary tables would help for the reader to understand the unique effects cannabinoids (e.g. THC, CBD) have on the immune system and viral infection. RESPONSE: Thank you, the graphic summary is now included.
  • Page 6, line 240: “Cannabis use does not adversely influence medication adherence/management among 240 adults living with HIV [52].” --> that is actually debated, different papers show otherwise, depending on how cannabis is used, medical or recreational. RESPONSE: We have now elaborated such differences.
  • The discussion is missing a more detailed review analysis on what the difference is between the diseases outlined and the actions of these various cannabinoids. 

Minor Comments: These errors are now corrected, thank you.

  • Page 2, line 68: “pro-inflammatory” --> should not have a hyphen; it is one word “proinflammatory”
  • Page 3, line 80: “… in turn, control other …” --> should read “… in turn, controls other …”
  • Page 5, lines 177-179: “Earlier research in humans and macaques suggested that cannabis may have an impact on plasma viral load. But later research suggests that cannabis use may be potentially beneficial in people with HIV infection.”  --> citations needed

Reviewer 2 Report

The main problems that I have with the review is:

a. Its lack of interest. There are many reviews summarizing the impact of cannabionoids on the immune system (with more infos and complete figures). The only and real novelty regarding your review would be the impact on cannabinoinds on cell infectivity. But, asit is indicated in line 171 (section Cannabis and infections), there is a paucity of data around this topic, which indicate that there are not enough infos in the literature to be able to mount such review.

b. There are no Tables and figures in this review. For me, it is unacceptable. When readers are interested by looking at one review, the latter has to summarize the info with figures.

c. Just 62 references for a review is pretty low. And the fact that 10 of them (representing 15% of total) are not even from our century (before 2000). Once again, for me, it is unacceptable since a review has to be timely with current observations.

Author Response

We respectfully disagree with the reviewer. It is unfortunate that the reviewer did not think the manuscript is acceptable for its importance, timeliness, and/or citing the work performed in ‘not in our century’. Respectfully we like to state that some of the pioneering work cited on cannabis, immune system, and infections was carried out in the 1980s and 1990s and that paved the way for current work cited in the manuscript. Incidentally, most Noble laureates began their work in the last century and received their Noble prize in this current century.

A graphic summary has been added.

Reviewer 3 Report

In this review manuscript, Maggirwar and Khalsa have elegantly reviewed the link between cannabis use, immune system, and viral infections. The authors used the NIH PubMed database and GoogleScholar by using keywords including cannabis, cannabinoids, tetrahydrocannabinol, THC, cannabidiol, CBD, marijuana, immune system, viral infections, HIV, HCV, and HTLV-I/II and wrote this review with updated literature.  A comment and not a critique is that the Inclusion of a schematic diagram for each section would strengthen the manuscript.

Comments:

  1. Include a section on future perspectives.
  2. Is there any information on the role of cannabis on inflammasome signaling?
  3. Discuss the role of cannabis on immunomodulation during viral infections in the context of mitochondria.
  4. Is anything known on the cannabis and gut-brain axis during viral infection?

Author Response

According to the reviewer #3, we have added the most relevant and available information on inflammasome and mitochondrial role in cannabis effects on immune system and viral infections. Most importantly and as correctly suggested, we have inserted a new section: Future Directions, where based on the identified gaps and our past decades of experience, we have suggested new avenues of research be explored on cannabis, immune function, and viral infections.    

Reviewer 4 Report

I enjoyed reading the review on Cannabis Use, Immune System, and Viral Infections by Drs. Maggirwar and Khalsa; two experts in the field. The review is focused and excellent and can be accepted as it is. A summary figure or any illustrations would further increase the impact of this manuscript.

Author Response

Thank you so much for encouraging comments.

Round 2

Reviewer 1 Report

The authors have addressed all my concerns and comments. Thank you. 

Reviewer 2 Report

revisions led to a better manuscript.